# Prevalence and predictors of use of long-term and short-acting reversible contraceptives among women of reproductive age in Wakiso and Hoima districts, Uganda: A cross-sectional study

Malachi Ochieng Arunda[1]*, Babirye Mary Estellah[1], Carl Fredrik Sjöland[1], Emmanuel Kyasanku[2], Stephen Mugamba[2], Vitalis Ofumbi Olwa[2], Robert Bulamba[2,3], Phillip Kato[2], James Nkale[2], Fred Nalugoda[2], Grace Nalwoga Kigozi[2], Gertrude Nakigozi[2], Godfrey Kigozi[2], Joseph Kagaayi[2,3], Deusdedit Kiwanuka[2], Stephen Watya[2], Anna Mia Ekström[1,4], Elin C. Larsson[1,5]

1 Department of Global Public Health, Karolinska Institutet, Stockholm, Sweden, 2 Africa Medical and Behavioural Sciences Organization (AMBSO), Hoima, Uganda, 3 School of Public Health, Makerere University, Kampala, Uganda, 4 Department of Infectious Diseases, South General Hospital, Stockholm, Sweden, 5 Department of Women's and Children's Health, Karolinska Institutet, Stockholm, Sweden

* malachi.ochieng.arunda@ki.se

**Data Availability Statement:** All data can be found in the manuscript and supporting information files.

## Abstract

Modern contraceptive use has increased globally, but unmet needs persist in low- and middle-income countries. This study in Uganda aimed to examine the prevalence and factors influencing the use of short-acting reversible contraceptives (SARC) like pills and long-term methods such as intrauterine devices. Limited evidence exists on the use of SARC and long-term methods in Uganda. Data from the Africa Medical and Behavioural Sciences Organization (AMBSO) Population Health Surveillance (APHS) in Hoima and Wakiso districts were analysed. Among the 1642 women aged 15–49 years, the prevalence of modern contraceptive use was 30% for SARC, and 18% for long-term method. Women with formal education were three times more likely to use long-term methods than those without formal education, relative risk ratios (RRR), 3.1–3.4, (95%CI 1.2–8.2). Joint decision-making for contraceptive use increased SARC usage, RRR 1.4 (95%CI 1.1–1.8). Urbanization played a role, with women in more urbanized Wakiso district less likely to use any modern contraception, RRR 0.6–0.7 (95%CI 0.5–0.9) compared to those living in the less urbanized Hoima. About half of the women in the study used modern contraceptives and the use of SARC was almost twice that of long-term methods. Increased access to contraception education for all women of reproductive age could significantly improve the use of long-term methods which offer more reliable protection against unintended pregnancies. The findings shed light on the need to strengthen both general and sexuality education to girls and women and to tailor contraception access for all in need, for mobile semi-urban as well as rural women. Well-informed strategies that engage young men and male partners in informed decision-making for contraceptive use could enhance progress.

**Funding:** The authors received no specific funding for this work.

**Competing interests:** The authors have declared that no competing interests exist.

## Introduction

Globally, in 2019, 44% of all women of reproductive age (15–49 years) used modern contraception methods. This constituted 91% of all contraceptive use worldwide, indicating only 9% use traditional contraception [1]. Although progress has been made globally, Africa registered the lowest contraceptive prevalence rate of 29% [1]. In 2020–2021, access to contraceptives was further affected by the COVID-19 pandemic in most low- and middle-income countries (LMIC) [2, 3]. Low contraception use can have far-reaching consequences, ranging from increased risk of unintended pregnancies and increased danger of health frailty caused by high parity to loss of productivity, poverty, and poor maternal and child health outcomes [4–6]. Investing in family planning would profoundly mitigate such risks [4, 7]. A study conducted in 172 countries using counterfactual modelling approach in 2012 found that satisfying family planning needs would reduce maternal deaths by 29% [4].

Contraceptive prevalence is defined as the proportion of women who are currently using or whose sexual partners are using *at least one* contraceptive method [8]. Contraceptive or family planning methods are sometimes divided into traditional and modern. Traditional methods include periodic abstinence (rhythm, lactational amenorrhea or calendar methods) and withdrawal. Modern methods are further categorized into short-acting reversible contraception (SARC) and long-term methods. Long-term contraception consists of both long-acting reversible contraception (LARC) such as intrauterine contraceptive devices (IUD) and permanent methods such as tubal ligation [9]. Low rates of contraceptive use indicate a gap between demand and access to sexual and reproductive health care services among women of reproductive age who would like to control childbearing, i.e., unmet need for contraception [10].

The challenge of unintended pregnancies and consequent unsafe abortions have persisted in many low- and middle-income countries (LMICs) [11–13]. Distal or indirect factors such sociodemographic factors, cultural norms and religious beliefs are critical determinants of contraceptive use through their linkages to direct barriers such as misconceptions, lack of information, low access, infrequent sex, and lack of access to preferred methods [10, 14–16].

Uganda, like other sub-Saharan African (SSA) countries, has made progress in expanding access to contraceptives, however, the unmet need for contraception is still very high (28%), and Uganda is among the countries with the lowest contraceptive prevalence in East Africa [17]. Studies conducted in LMICs have broadly focused on investigating the prevalence of general modern contraceptive use [18–30]. Very few studies in Uganda have examined the prevalence and factors associated with use of short-acting and long-term contraceptive methods [31, 32]. A 2016 study by Dau et al. using nationally representative data indicated that 26% of women aged 15–49 used long-acting reversible contraceptives (LARC) in Uganda [32]. Further, qualitative studies conducted in Uganda have explored phenomena of knowledge and change of attitudes based on very few participants and thus may not be generalizable to majority of Ugandans [33]. However, the predictors of availability and accessibility of contraception methods are dynamic and complex in LMIC [19], requiring constant investigations to determine prevalence, trends and patterns. There is a need to examine the use of SARC and long-term methods in rural, semi-urban and urban settings in LMIC to reveal differential contraceptive needs. Thus, the aim of this study is to determine the prevalence and factors associated with the use of SARC and long-term methods among women of reproductive age (15–49 years) in rural, semi-urban and urban areas of Wakiso and Hoima districts in Uganda. This may contribute to extend knowledge, improve understanding, and inform efforts and strategies aimed at improving effective contraceptive use in similar LMIC settings.

### Conceptual framework

Research evidence indicate that theory-based studies are critical in determining care-seeking behaviours and utilization of health services including the uptake of modern contraceptive methods. In this study, we adapted Anderson´s and Newman´s behavioural model of utilization of health services [34]. Key elements in the model include predisposing, enabling and need (modified need) factors which motivate, or hinder uptake and sustained use of a given contraception method. Predisposing factors consist of; demographic factors such as region, age, education, enabling factors such as support for decision making and type of residential area, and need-for-contraception factors that include conditions, sexual/reproductive status such as being married or sexually active, and parity that may require contraceptive use [27, 34–37].

## Materials and methods

### Study design and setting

The study obtained cross-sectional data from the on-going Africa Medical and Behavioural Sciences Organization (AMBSO)´s Population Health Surveillance (APHS), which is an open longitudinal population-based cohort study established in 2017 in Wakiso and Hoima districts in Uganda [38]. Wakiso district is in the central region of Uganda surrounding Kampala city, has a rapidly growing population of about 3.1 million (2020) [39] and is more urbanized (than Hoima). Hoima district on the other hand is situated in western Uganda with a population of about 0.4 million [39]. The APHS study design and data collection procedures are detail in a cohort description elsewhere [38]. Both districts constitute urban, semi-urban and rural communities from which the study sample was collected. The APHS cohort consists of individuals aged 13 years and above living in Wakiso and Hoima districts. A community mapping exercise (involving local stakeholders) was performed to ensure the communities selected for data collection are representative of the different community types in Uganda. The survey collects data on a wide range of health and behavioural topics including sexual and reproductive health, contraceptive use, nutrition, determinants of health, communicable and non-communicable diseases among others.

### Study participants and data collection

A baseline census was conducted in 2018–2019 that identified 10,929 individuals aged 13 years and above who were eligible for inclusion in the study. In this study, we use data from a follow-up survey (APHS2) that was conducted between January-March, and August-December 2020 in Hoima and Wakiso districts. A total of 4276 individuals (56% females and 44% females) aged 13–82 years successfully participated in the survey giving a response rate of 39.1%. Out of the 4276 individuals, we selected 1642 females aged 15–49 years who reported ever having had a sexual partner and were not pregnant at the time of data collection for this current study and after eliminating seven females who were non-respondents on contraceptive use, a total of 1635 females were included in the analysis. In adherence to the terms of the ethical approval and informed consent, data for variables related to the manuscript can be availed upon sending a formal request to AMBSO through the corresponding author. Additionally, we have made available, part of the output of the multinomial logistic regression analysis that was conducted for model 3 of Table 2 in the manuscript.

### Variables

**Outcome variable.**   The primary outcome was current use of contraception, assessed through the question "*Are you and your partner currently using any of the following family*

*planning methods*?*"*. Participants could respond by selecting one or more methods which were classified in three main categories namely; short-acting reversible (SARC), long-term methods (LARC) and permanent methods) and traditional/non-use as shown in the Box 1 below [9].

---

### Box 1. Classification of contraceptive types used by women in Uganda

| Short-acting reversible contraception | Long-term methods | Traditional methods/non-use |
|---|---|---|
| Pills, condoms, injectables (including sayana), spermicides, cervical cap. | Long-acting methods: Norplant, intrauterine contraceptive device, and permanent methods: tubal ligation, vasectomy | Herbs/traditional medicine Withdrawals, breast-feeding, rhythm/calendar, abstinence, all other non-contraceptive users |

---

## Independent variables

Among sociodemographic variables, age was reported as a continuous variable but categorized into 15–20 and 21–49, place of residence (semi-urban, rural, and urban). Parity was categorized as (1–3, 4–15, never pregnant/no child). Religion was made into a categorical variable (Catholic, Protestants, Muslims, none/other Christians). Education was further categorized into, no schooling, any primary, any secondary, or higher education. Marital status was made into binary variable: 'yes´ to mean married and 'No´ for non-married. Age at first sex was reported as a continuous variable and categorized as (≤14, 15–19, 20–42, and don't remember), age at first marriage was classified in five ways (≤14, 15–19, 20–42, and never been married), age at first birth was grouped into (≤14, 15–19, 20–38, and never been pregnant), contraceptive decision making was classified into (mainly respondent, mainly husband, joint decision). Sexual activity was assessed by the following question: *Have you had sexual intercourse with any person in the last 12 months*? The response was dichotomized into yes or no.

## Statistical analysis

**Data analysis.** The data was analysed using STATA version 16 [40]. Cross-tabulations were used to examine the distribution of participants´ demographic, sexual, and reproductive characteristics, and variable categories across contraceptive types. Backward stepwise selection was used to identify key variables fit for inclusion in the multinomial logistic regression model, using a cut-off p-value of 0.2 to eliminate non-associated variables [41, 42]. Multinomial logistic regression models examined the associations between independent across SARC and long-term contraception methods, with traditional method/non-use as the base category. Model 1 included demographic variables (predisposing factors such as age and education) while model 2 included both predisposing and enabling variables such as type of residential area while final model (model 3) included all the variables in predisposing factors models 1 and 2 in addition to need factors such as parity or being sexually active. All these factors are discussed in the conceptual framework in the introduction section. The independent variables mutually adjusted for each other. The models estimated the relative risk ratios (RRR) with 95% confidence intervals (CI).

**Missingness.** Missingness was minimal (less than 5%) and negligible for most of the independent and outcome variables (0.43%). This study is part of a known cohort and thus where

some data was missing on variable *religion*, we assigned the religious affiliation they had stated in the baseline survey. This return-to-baseline imputation method is known to generate very minimal or no bias or variance [43], especially since religious affiliations may not normally change rapidly.

**Patients and public involvement.** The public was only involved as research participants. They were not directly involved in the design, or recruitment, conduct, reporting, or dissemination plans of this study.

## Ethical considerations

The study was approved by Clark International University (CIU) Research Ethics Committee (REC), the local Institution Review Board and cleared by the Uganda National Council for Science and Technology (SS4468). Before data collection, written informed consent was obtained from all participants and participation was voluntary. For each of the participants under 18 years of age in the study, a written informed consent was obtained from the parent/guardian. Participants were assigned identification numbers for confidentiality and anonymity. There were no substantive risks for participating in the study. Further, interviewers were trained in Good Clinical Practice (GCP) were equipped with counselling skills to support participants who may have experienced distress as a result of participating in the survey.

## Results

### Description of study population across contraception methods

Fig 1 shows the types of contraceptive methods or non-use among women in the study. About 30% and 18% of the 1635 women in the study used SARC or long-term contraception methods respectively, while only 6% used traditional methods and the rest (46%) were non-contraceptive users (i.e., 52% non-modern contraceptive users).

Table 1 below shows the distribution of women's background and sexual characteristics by type of contraception use versus non-use. About 90% of women had either primary or secondary education and there were minimal proportional differences in the use of SARC, long-term contraception or traditional methods/non-use across women's various educational levels. Over 45% of the women were already married at and had given birth at the age of 19 and among those using modern contraception, about 90% had their first sexual encounter at the age of 19 or earlier. A similar proportion (85%) of those who had their first sexual encounter before age 19 were in the non-use/traditional method category. A higher proportion of women (54%) who used SARC or long-term methods made joint decisions with their partners while majority (55.6%) of women in the traditional/non-user categories had not consulted a partner.

Table 2 and Fig 2 present the findings of the multinomial regression analysis for the associations between independent factors and the use of SARC and long-term methods, with the traditional methods/non-use as the base category. Women living in Wakiso district were about 30–40% less likely to use either SARC and long-term methods compared to those living in Hoima district, the RRRs ranged from 0.6 to 0.7 (95% CI 0.5 to 0.9) in models 2 and final model (model 3). Generally, women with primary or secondary/higher education were more likely to use long-term methods than those with no formal schooling, final model (model 3) including all confounding factors, RRRs ranged from 3.1–3.4 (95% CI 1.2 to 8.2). The corresponding associations for using SARC methods were not statistically significant.

Women living in semi-urban areas were about 60% more likely to use long-term contraception methods compared to those in the rural, RRR 1.6 (95% CI 1.1–2.1) in the final model. Living in the urban compared to rural was not associated with use of either SARC or long- term methods. However, Women living in semi-urban and those having 1–3 live births (verses

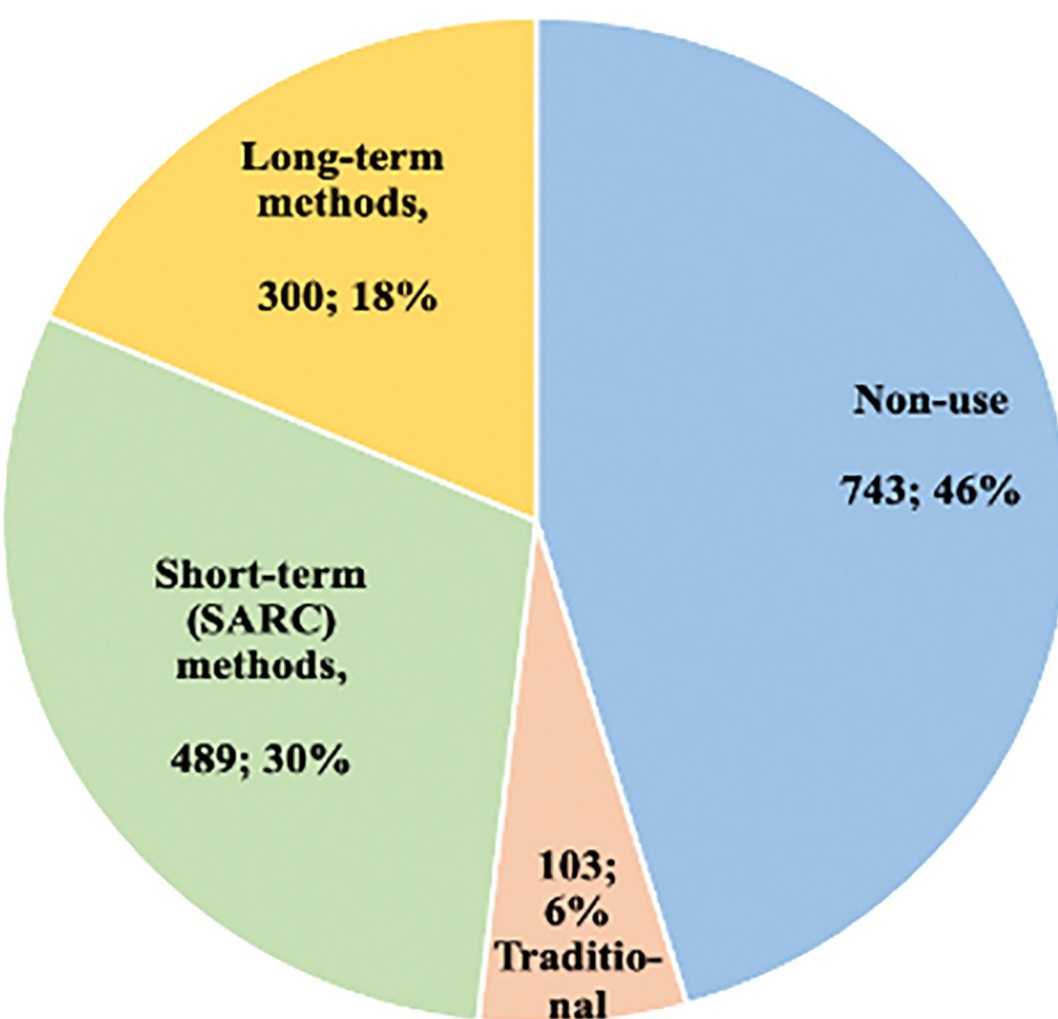

**Fig 1. Number and proportions of women using long-term, SARC, traditional contraception methods and non-users in Wakiso and Hoima districts in Uganda, 2020.**

those never pregnant) also indicated higher likelihood of using SARC methods, nonetheless these associations were borderline not statistically significant, RRR 1.4 (95%CI 1.0–1.9) and RRR 1.8 (95%CI 0.9–3.9).

Joint (respondent and husband) decision-making in the use of contraceptives could have motivated the use of SARC methods when compared with the respondent woman making independent decisions to use contraceptive, RRR 1.4 (95%CI 1.1–1.8) (final model). Corresponding associations with long-term methods were borderline significant in the full model (model 3), RRR 1.3 (95%CI 0.9–1.7), although in model 2 (adjusting for predisposing factors including demographic factors such as age and education and enabling factors such as support for decision making and type of residential area) the association was statistically significant. Being married significantly lowered the likelihood of using SARC methods compared to being single, RRR 0.7 (95%CI 0.5–0.9). Instead, married women were more likely to use long-term methods although borderline significant, RRR 1.4 (95%CI 0.9–2.1).

Women who had been sexually active in the past 12 months were 16 times more likely to use SARC and 6 times more likely to use long-term methods, RRR 5.8 to 16.3 (95%CI 2.9 to

**Table 1. Distribution of socio-demographic characteristics, sexual activity, and decision-making variables by type of contraceptive use among 1635 women in Wakiso and Hoima districts in Uganda in 2020.**

| Variable | Variable categories | Non-use/ traditional methods | | Short-Acting Reversible Contraception (SARC) | | Long-term Contraception | |
|---|---|---|---|---|---|---|---|
| | | n = 846 | (%) | n = 489 | (%) | n = 300 | (%) |
| Age | 15–20 | 102 | 12.6 | 71 | 14.5 | 25 | 8.3 |
| | 21–49 | 744 | 87.9 | 418 | 85.5 | 275 | 91.7 |
| Region of residence | Hoima | 430 | 50.8 | 278 | 56.9 | 170 | 56.7 |
| | Wakiso | 416 | 49.1 | 211 | 43.2 | 130 | 43.3 |
| Place of residence | Semi-Urban | 241 | 28.5 | 144 | 29.5 | 117 | 39.0 |
| | Rural | 264 | 31.2 | 130 | 26.6 | 97 | 32.3 |
| | Urban | 341 | 40.3 | 215 | 43.9 | 86 | 28.7 |
| Number of live births | 0–3 | 416 | 49.2 | 298 | 60.9 | 177 | 59.0 |
| | 4–17 | 320 | 37.8 | 135 | 27.6 | 120 | 40.0 |
| | Never pregnant | 110 | 13.0 | 56 | 11.5 | 3 | 1.0 |
| Religion | Catholic | 332 | 39.2 | 187 | 38.2 | 111 | 37.1 |
| | Protestant | 252 | 29.8 | 159 | 32.5 | 97 | 32.3 |
| | Muslim | 118 | 13.9 | 67 | 13.7 | 37 | 12.3 |
| | None/other Christians | 144 | 17.0 | 76 | 15.5 | 55 | 18.3 |
| Attained level of education | No schooling | 40 | 4.7 | 16 | 3.3 | 6 | 2.0 |
| | Any primary | 397 | 46.9 | 202 | 41.3 | 152 | 50.7 |
| | Any secondary | 361 | 42.7 | 237 | 48.5 | 133 | 44.3 |
| | Higher | 48 | 5.7 | 34 | 7.0 | 9 | 3.0 |
| Marital status | Married | 501 | 59.2 | 298 | 60.9 | 230 | 76.7 |
| | Single | 345 | 40.8 | 191 | 39.1 | 70 | 23.3 |
| Age at first marriage (years) | ≤ 14 | 35 | 4.1 | 16 | 3.3 | 19 | 6.3 |
| | 15–19 | 349 | 41.2 | 206 | 42.1 | 141 | 47.0 |
| | 20–42 | 317 | 37.5 | 172 | 35.2 | 111 | 37.0 |
| | Never married | 145 | 17.1 | 95 | 19.4 | 29 | 9.7 |
| Age at first sex | ≤ 14 | 131 | 15.9 | 74 | 15.4 | 44 | 15.0 |
| (years) | 15–19 | 573 | 69.8 | 347 | 72.3 | 217 | 74.1 |
| | ≥20 | 117 | 14.3 | 59 | 12.3 | 32 | 10.9 |
| Age at first birth | ≤ 14 | 38 | 4.5 | 20 | 4.1 | 20 | 6.7 |
| (years) | 15–19 | 374 | 44.2 | 216 | 44.2 | 164 | 54.7 |
| | 20–38 | 305 | 36.1 | 188 | 38.4 | 113 | 37.7 |
| | Never pregnant | 129 | 15.2 | 65 | 13.3 | 3 | 1.0 |
| Decision to use contraceptives | Mainly respondent | 457 | 55.6 | 208 | 42.5 | 132 | 44.0 |
| | Mainly husband | 29 | 3.5 | 18 | 3.7 | 5 | 1.7 |
| | Joint decision | 336 | 40.9 | 263 | 53.8 | 163 | 54.3 |
| Sexually active in past 12 months | Yes | 680 | 80.4 | 481 | 98.4 | 291 | 97.0 |
| | No | 166 | 19.6 | 8 | 1.6 | 9 | 3.0 |

36.2), compared to those sexually inactive in the past 12 months. Age at first sex was not associated with the use of either SARC or long-term contraception.

## Discussion

Overall, 52% of women 15–49 years of age in this Ugandan population cohort who reported to ever have had sexual partners, were either non-users of contraception or used traditional methods, indicating high unmet need for both SARC and long-term contraception in Uganda.

**Table 2. Multinomial logistic regression showing relative risk ratios (RRR, 95% confidence interval (CI)) for the associations between selected independent variables and short-acting and contraceptive use across short -and long-term contraception use methods among 1635 women of reproductive age in Wakiso and Hoima districts in Uganda in 2020.**

| | Base category-Traditional/non-users | Model 1 | | Model 2 | | Final model (Model 3) | |
|---|---|---|---|---|---|---|---|
| | | Short-acting reversible contraception (SARC) | Long-term methods | Short-acting reversible contraception (SARC) | Long-term methods | Short-acting reversible contraception (SARC) | Long-term methods |
| **Region** | Hoima | Ref. | Ref. | Ref. | Ref. | Ref. | Ref. |
| | Wakiso | 0.8 (0.6–1.0) | 0.7 (0.6–1.0) | 0.7 (0.5–0.9) * | 0.6(0.5–0.8) * | 0.7(0.5–0.9) * | 0.6 (0.5–0.8) * |
| **Attained level of education** | No schooling | Ref. | Ref. | Ref. | Ref. | Ref. | Ref. |
| | Any primary | 1.2(0.7–2.2) | 2.6(1.1–6.3) * | 1.2(0.7–2.3) | 2.9(1.2–7.0) * | 1.1(0.6–2.1) | 3.1(1.2–7.6) * |
| | ≥secondary | 1.6(0.9–3.0) | 2.6(1.1–6.3) * | 1.5(0.8–2.9) | 3.1(1.3–7.6) * | 1.3(0.7–2.6) | 3.3(1.3–8.2) * |
| **Woman's age (years)** | 15–20 | Ref. | Ref. | Ref. | Ref. | Ref. | Ref. |
| | 21–49 | 0.6(0.3–0.9) * | 0.5(0.3–0.9) * | 0.6(0.4–0.9) * | 0.6(0.3–1.0) | 0.9 (0.5–1.5) | 0.8(0.4–1.4) |
| **Place of residence** | Rural | | | Ref. | Ref. | Ref. | Ref. |
| | Semi-urban | | | 1.4 (1.1–2.0) * | 1.6(1.1–2.2) * | 1.4 (1.0–1.9) | 1.6(1.1–2.2) * |
| | Urban | | | 1.3(1.0–1.8) | 0.7 (0.5–1.0) | 1.3 (1.0–1.8) | 0.7(0.5–1.1) |
| **Decision to use contraceptives** | Mainly respondent | | | Ref. | Ref. | Ref. | Ref. |
| | Mainly husband | | | 1.8(0.8–3.2) | 0.8(0.3–2.3) | 1.8(0.8–3.8) | 0.9(0.3–2.6) |
| | Joint decision | | | 1.5(1.2–2.0) * | 1.4(1.1–1.9) * | 1.4(1.1–1.8) * | 1.3(0.9–1.7) |
| **Marital status** | Single | | | | | Ref. | Ref. |
| | Married | | | | | 0.7 (0.5–0.9) * | 1.4(0.9–2.1) |
| **Age at first marriage (years)** | ≤ 14 | | | | | Ref. | Ref. |
| | 15–19 | | | | | 1.0(0.5–2.0) | 0.5(0.2–1.0) |
| | 20–42 | | | | | 0.8(0.4–1.7) | 0.4(0.2–0.9) * |
| | Never married | | | | | 1.0(0.4–2.4) | 0.9(0.3–2.3) |
| **Age at first sex (years)** | ≤ 14 | | | | | Ref. | Ref. |
| | 15–19 | | | | | 1.1(0.7–1.6) | 1.5(0.9–2.4) |
| | 20–30 | | | | | 0.9(0.5–1.5) | 1.1(0.6–2.2) |
| **Number of live births** | Never pregnant | | | | | Ref. | Ref. |
| | 1–3 | | | | | 1.8 (0.9–3.9) | – |
| | 4–15 | | | | | 1.2 (0.5–2.6) | – |
| **Sexual activity in the past 12 months** | No | | | | | Ref. | Ref. |
| | Yes | | | | | 16.3(7.3–36.2) * | 5.8(2.9–12.5) * |

Among those using modern methods, the use of SARC was almost twice that of long-term methods that normally are considered safe in terms of protecting against unwanted pregnancy. Overall, having primary or secondary/higher education tripled the likelihood to use long-term methods, compared to women with no schooling at all, making education one of the most important demographic predictors of long-term contraceptive use in this population.

Interestingly, living in a semi-urban area, was associated with the use of long-term methods, while living in the more urban Wakiso district decreased the likelihood of using modern contraception compared the more rural Hoima district. This may be explained by higher mobility among semi-urban women, possibly decreasing access to family planning or contraception

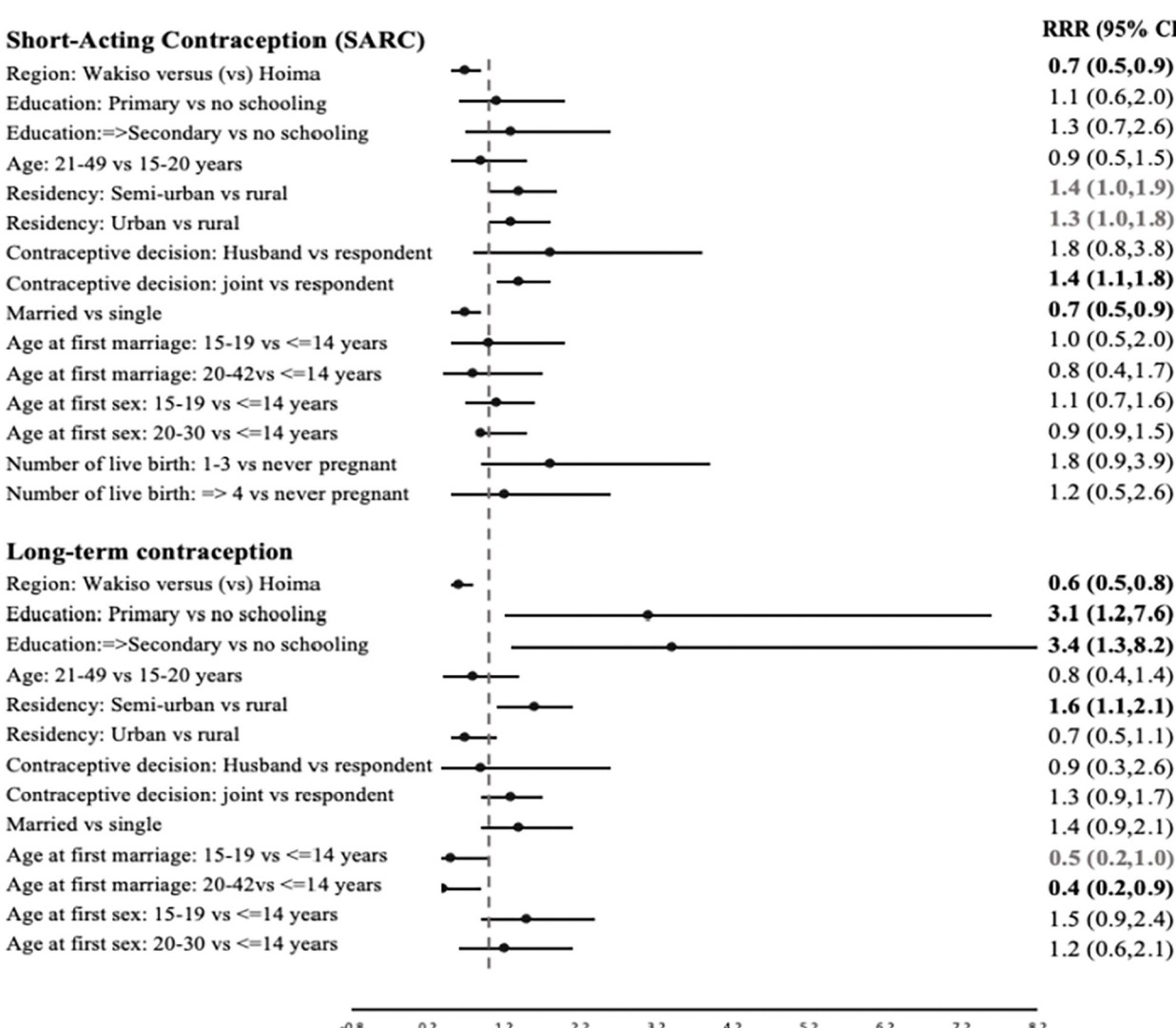

**Fig 2. Forest plot summary of model 3 (Table 2) showing relative risk ratios (RRR 95%CI) for the associations between independent variables and contraceptive use across SARC and long-term methods in Wakiso and Hoima districts in Uganda, 2020.**

services. However, further research is needed to ascertain this. Being sexually active in the past 12 months was the strongest overall factor associated with increased use of both SARC and long-term contraception.

The combined prevalence of SARC and long-term contraceptive use (48%) in this Ugandan urban/semi-urban and rural cohort of women of reproductive age, is comparable to previous findings from Wakiso district (47%) with data collected in 2019 [22]. This implies that the Ministry of Health's vision to reduce the unmet need for contraception to 10% by 2020 was not achieved [44]. Similarly, the contraceptive prevalence rate in our study is much lower than the international target of 75% [45]. This signals gaps in access, and a demand for contraception that must be addressed to meet the 2030 projections [46]. Considering that Uganda had a harsh and extended COVID-19-related mobility restrictions affecting access to health care as well as information and prevention to young women due to the 2-year long school closure, the situation may have been exacerbated, indicating the need to scale -up investment in family planning services and education for young women [21].

Earlier studies from sub-Saharan Africa including Uganda concur with our findings of a higher use of SARC than of long-term methods [22, 37, 47]. A study conducted in selected health centers in Uganda revealed that SARC methods can be more easily accessed, obtained at lower costs with a higher perceived privacy and fewer perceived side effects [48]. Further, SARC do not require a health worker for monitoring, and women can discontinue usage at their own convenience [48]. These factors possibly explain the preferences for SARC methods observed in our cohort. However, SARC require regular refills, which in turn are more frequently affected by logistic challenges, thus the effectiveness relies on both access, compliance and adherence. Thus, the use of SARC methods has been associated with high discontinuation rates [48, 49]. In comparison, permanent and long-acting reversible methods offer effective protection against unplanned pregnancies for longer periods, enabling women to control their family size, delay and space children without need for resupply [37, 50, 51]. This may partly explain the strong associations between semi-urban residency and long-term contraception methods. Women in semi-urban areas may have lower access to contraceptives compared to their urban counterparts, and thus may opt for long-term methods rather than SARC.

Residing in Wakiso compared to Hoima was also associated with 30–40% lower use of both SARC and long-term methods. A possible explanation is that Wakiso has almost five times the population of Hoima and has a huge proportion of urban populations living in informal settlements (slums), characterized by poverty, poor health care facilities, which limit access and utilization of family planning services [22, 52, 53]. This also partly explains why, despite better proximity to health services, living in an urban area, compared to a rural, did not indicate a significant association with long-term methods.

More than half of women using modern contraception indicated that this was a joint decision made by the couples, implying an understanding of its importance and a willingness for family planning by both partners. Married women indicated lower odds of using SARC and tendency towards use of long-term methods, suggesting a mutual willingness of couples to use effective family planning methods, in line with earlier findings from Tsui et al. across SSA [54]. Studies in SSA show that male partners play a key role in the choice of modern contraception and depending on their knowledge level, cultural inclinations and perceptions, they can influence usage both negatively or positively [37, 55]. In this study, male partners negatively influenced the use of SARC methods. A considerable proportion of husbands tend to influence their wives to opt out of or stop modern contraceptive use [44], and this is easier done when short-term SARC methods are used that does not require provider interventions.

It is known that formally educated women also tend to marry educated men and partly, in congruence with our findings, studies show that higher education for both males and females increases informed decisions for long-term contraceptive use [19, 48, 56]. investing in education for the general population and ensuring access to comprehensive reproductive health education, particularly for young women and girls, could profoundly improve effective use of modern contraceptives and uptake of long-acting methods.

According to the Ugandan National Strategic Plan 2015–2025, the Ministry of Health is committed to improving family planning through various strategies such as ensuring the availability of contraceptives, capacity building of service providers, scaling up of youth-friendly services etc [44, 57, 58]. However, as in many sub-Saharan African countries, most women, especially young ones, in Uganda still face significant health system, policy and socio-cultural (stigma) barriers that profoundly hinder their use of contraceptives especially long-term methods [25, 59, 60].

### Strengths and limitations

The selection of sample populations from rural, urban, and semi-urban areas across the two districts, in addition to the relatively large sample size should provide a fair reflection of Ugandan demography making our findings generalizable across the country. A limitation in this study is that, although the cohort is longitudinal, the data was collected through a cross-sectional survey and recall bias cannot be completely eliminated. Further, we were unfortunately unable to examine some key factors that are known to affect contraceptive use such as misconceptions, contraceptive awareness and attitudes, and this could have limited our insight into the challenges for use of SARC and long-term contraceptive methods.

### Conclusion

We found that only about half of the women in this Ugandan population cohort use modern contraceptive methods and the use of short-term (SARC) methods was almost twice that of long-term contraception. Education, living in semi-urban and joint decision making for contraceptive use were all associated with long-term contraception and more than half of women on modern contraception had made a joint decision with their partners. These findings suggest a need to strengthen both general education and sexuality education to girls and women of reproductive age that includes contraception use, counselling and access. This could significantly improve the uptake of modern contraception especially long-term methods which offer reliable protection against unintended pregnancies. Well-informed strategies that engage young men and male partners in positive and informed decision-making for contraceptive use could also contribute to progress.

### Supporting information

**S1 Data. Dataset.**
(DTA)

**S1 Text. List of legends.**
(DOCX)

### Acknowledgments

Much gratitude to the APHS participants and community leaders and volunteers, ABMSO community Advisory Board, Wakiso and Hoima district directorates of Health Services and AMBSO staff for their support.

### Author Contributions

**Conceptualization:** Babirye Mary Estellah, Anna Mia Ekström, Elin C. Larsson.

**Data curation:** Carl Fredrik Sjöland, Emmanuel Kyasanku, Stephen Mugamba, Robert Bulamba.

**Formal analysis:** Malachi Ochieng Arunda, Babirye Mary Estellah.

**Funding acquisition:** Fred Nalugoda, Anna Mia Ekström, Elin C. Larsson.

**Investigation:** Malachi Ochieng Arunda, Babirye Mary Estellah, Anna Mia Ekström, Elin C. Larsson.

**Methodology:** Malachi Ochieng Arunda, Babirye Mary Estellah, Carl Fredrik Sjöland, Fred Nalugoda, Gertrude Nakigozi, Godfrey Kigozi, Anna Mia Ekström, Elin C. Larsson.

**Project administration:** Vitalis Ofumbi Olwa, Robert Bulamba, Phillip Kato, James Nkale, Fred Nalugoda, Grace Nalwoga Kigozi, Gertrude Nakigozi, Godfrey Kigozi, Joseph Kagaayi, Deusdedit Kiwanuka, Stephen Watya, Anna Mia Ekström, Elin C. Larsson.

**Resources:** Anna Mia Ekström, Elin C. Larsson.

**Supervision:** Fred Nalugoda, Anna Mia Ekström, Elin C. Larsson.

**Validation:** Emmanuel Kyasanku, Stephen Mugamba, Vitalis Ofumbi Olwa, Robert Bulamba, Phillip Kato, James Nkale, Fred Nalugoda, Grace Nalwoga Kigozi, Gertrude Nakigozi, Godfrey Kigozi, Joseph Kagaayi, Deusdedit Kiwanuka, Stephen Watya, Anna Mia Ekström, Elin C. Larsson.

**Writing – original draft:** Malachi Ochieng Arunda, Babirye Mary Estellah.

**Writing – review & editing:** Malachi Ochieng Arunda, Carl Fredrik Sjöland, Emmanuel Kyasanku, Stephen Mugamba, Anna Mia Ekström, Elin C. Larsson.

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
