## [Decision Letter · Decision Letter 0]

4 May 2023

PGPH-D-22-01907

Prevalence and predictors of long-term and short-acting reversible contraception use among women of reproductive age in Wakiso and Hoima districts, Uganda: a cross-sectional study

Dear Dr. Arunda,

Thank you for submitting your manuscript to PLOS Global Public Health. After careful consideration, we feel that it has merit but does not fully meet PLOS Global Public Health’s publication criteria as it currently stands. Therefore, we invite you to submit a revised version of the manuscript that addresses the points raised during the review process.

Please note that we have only been able to secure a single reviewer to assess your manuscript. We are issuing a decision on your manuscript at this point to prevent further delays in the evaluation of your manuscript. Please be aware that the editor who handles your revised manuscript might find it necessary to invite additional reviewers to assess this work once the revised manuscript is submitted. However, we will aim to proceed on the basis of this single review if possible.  

We look forward to receiving your revised manuscript.

Kind regards,

Dario Ummarino, PhD

Staff Editor

Journal Requirements:

Additional Editor Comments (if provided):

Reviewers' comments:

Reviewer's Responses to Questions

**Comments to the Author**

1. Does this manuscript meet PLOS Global Public Health’s publication criteria? Is the manuscript technically sound, and do the data support the conclusions? The manuscript must describe methodologically and ethically rigorous research with conclusions that are appropriately drawn based on the data presented.

Reviewer #1: Yes

2. Has the statistical analysis been performed appropriately and rigorously?

Reviewer #1: Yes

3. Have the authors made all data underlying the findings in their manuscript fully available (please refer to the Data Availability Statement at the start of the manuscript PDF file)?

Reviewer #1: Yes

4. Is the manuscript presented in an intelligible fashion and written in standard English?

Reviewer #1: Yes

5. Review Comments to the Author

Reviewer #1: Overall, this is a well-written manuscript on a highly relevant and useful topic. Please see my comments and questions in the attached version of your manuscript. Many of them are minor suggestions. Please review my comments on your conclusions section. It is unclear how you reached the conclusion that expanded family planning clinics and mobile clinics are needed from your data analysis. What conclusions can you reach strictly from this analysis?

6. PLOS authors have the option to publish the peer review history of their article (what does this mean?). If published, this will include your full peer review and any attached files.

**Do you want your identity to be public for this peer review?** For information about this choice, including consent withdrawal, please see our Privacy Policy.

Reviewer #1: **Yes: **Laura Hoemeke

---

## [Decision Letter · Decision Letter 1]

13 Nov 2023

Prevalence and predictors of use of long-term and short-acting reversible contraceptives among women of reproductive age in Wakiso and Hoima districts, Uganda: a cross-sectional study

PGPH-D-22-01907R1

Dear Dr Arunda,

We are pleased to inform you that your manuscript 'Prevalence and predictors of use of long-term and short-acting reversible contraceptives among women of reproductive age in Wakiso and Hoima districts, Uganda: a cross-sectional study' has been provisionally accepted for publication in PLOS Global Public Health.

Best regards,

Anushka Ataullahjan

Guest Editor

Please address the minor comments highlighted by reviewer before publication, thank you!

Reviewer Comments (if any, and for reference):

Reviewer's Responses to Questions

**Comments to the Author**

1. If the authors have adequately addressed your comments raised in a previous round of review and you feel that this manuscript is now acceptable for publication, you may indicate that here to bypass the “Comments to the Author” section, enter your conflict of interest statement in the “Confidential to Editor” section, and submit your "Accept" recommendation.

Reviewer #1: All comments have been addressed

Reviewer #2: (No Response)

2. Does this manuscript meet PLOS Global Public Health’s publication criteria? Is the manuscript technically sound, and do the data support the conclusions? The manuscript must describe methodologically and ethically rigorous research with conclusions that are appropriately drawn based on the data presented.

Reviewer #1: Yes

Reviewer #2: Yes

3. Has the statistical analysis been performed appropriately and rigorously?

Reviewer #1: Yes

Reviewer #2: No

4. Have the authors made all data underlying the findings in their manuscript fully available (please refer to the Data Availability Statement at the start of the manuscript PDF file)?

Reviewer #1: Yes

Reviewer #2: Yes

5. Is the manuscript presented in an intelligible fashion and written in standard English?

Reviewer #1: Yes

Reviewer #2: Yes

6. Review Comments to the Author

Reviewer #1: Comments on the original manuscript have been addressed.

Reviewer #2: REVIEW: Prevalence and predictors of use of long-term and short-acting reversible 4 contraceptives among women of reproductive age in Wakiso and Hoima districts, 5 Uganda: a cross-sectional study

In the abstract section, in the results the authors should include specific predictors of contraceptive use identified in this study.

Financial disclosure: check spelling “Institute”.

Include keywords in abstract.

Introduction: Your first sentence is not clear.

Line 166: “which is”…seems missing

Study design and setting: Important details such as sampling techniques, the type of questionnaire used should be included.

Study participants and data collection: What is the rational of selecting the number of participants out of the total participants in the census.

Line 202 what do you mean by “Responses were multiple choice”

Line 250: what is the relevant of this section to this paper? “the patient and public involvement”.

Results: This study objectives is to determine prevalence as well as predictors of contraceptive use, hence the results should be able to clearly identify predictors that are statistically significant in this study. Similar to the discussion of results in the discussion section.

7. PLOS authors have the option to publish the peer review history of their article (what does this mean?). If published, this will include your full peer review and any attached files.

**Do you want your identity to be public for this peer review?** For information about this choice, including consent withdrawal, please see our Privacy Policy.

Reviewer #1: No

Reviewer #2: **Yes: **Queen Esther Adeyemo
